# Service quality assessment and enhancement using Kano model

**Sharareh Kermanshachi**[1]*, **Thahomina Jahan Nipa**[1], **Halil Nadiri**[2]

**1** Department of Civil Engineering, University of Texas at Arlington, Arlington, Texas, United States of America, **2** Department of Economics and Administrative Sciences, Cyprus International University, Nicosia, Republic of Cyprus

* sharareh.kermanshachi@uta.edu

**Data Availability Statement:** All relevant data is included in the paper and Supporting Information files.

**Funding:** The author(s) received no specific funding for this work.

## Abstract

Success in the retail sector is highly dependent on customer satisfaction. Maintaining a competitive edge depends upon the service providers knowing and enacting what is important to their customers. Multiple studies have employed various research approaches to identifying characteristics of customer satisfaction in different sectors as well as retail sector. However, very few have determined such characteristics using multiple approaches simultaneously in the retail store. This study aims to identify, categorize, and rank the retail store attributes, based on their effects on customer satisfaction. A survey focusing on retail store characteristics that impact customer satisfaction was developed and distributed. Over 400 responses were collected and evaluated, using the Kano model. Results showed that visually appealing facilities and error-free transactions are of prime importance to customers. They are taken for granted, but their absence plays a significant role in customer dissatisfaction. An easy-to-navigate store layout and readily available service increase customer satisfaction, but their absence doesn't decrease customer satisfaction. Clean public areas and modern-looking equipment are important, and improvements to them increase customer satisfaction at a proportional rate. The findings of this study will assist service providers in realizing the relative importance of the attributes of retail stores and in evaluating the impacts of their current practices on customer satisfaction levels. Such importance will also help retail sector policy makers in mandating policies focusing on must-have attributes to preserve customer satisfaction.

## Introduction

The retail trade is experiencing significant growth in the present economy. In the US, its annual gross output was approximately $2 trillion in 2018 [1], and in 2016, it employed more than ten percent of the total work force [2]. The many opportunities available in the retail sector are leading more and more people to invest and/or to engage in it, which increases the competition and makes it vital that service providers think and act innovatively to keep their customers happy. Customer satisfaction is considered a measure of customer loyalty [3, 4], and customer loyalty produces commitment and attachment that prevent the customer from

**Competing interests:** The authors have declared that no competing interests exist.

exploring the advantages of competitors [5, 6]. Quality service determines whether the store will have a loyal and satisfied customer base, but modern customers demand more than just a high-quality product. They have an increased awareness of intangible services [7], and desire an amiable shopping experience in addition to efficient basic services [8]. This presents a challenge for retailers, as the differences in shoppers make it difficult to identify what exactly constitutes a satisfying, agreeable experience.

To enjoy an advantage over its competitors, a service provider must know the attributes that contribute to their customers' satisfaction. Several studies have employed various research approaches to identifying customer satisfaction characteristics in different sectors as well as the retail sector. Kim et al. [9] studied customer equity and customer satisfaction in traditional and new retail formats using regression analysis. In another study, Nicod et al [10] found that providing proactive training will increase the sales value per customer but will not enhance customer satisfaction. Veloso et al. [11] established using a multi-level and hierarchical model that the customer satisfaction and service perceived quality has a significant correlation among them with respect to the retail industry. The above-mentioned studies have a limited scope in a prioritized list of characteristics to focus on when enhancing customer satisfaction in the retail store. Moreover, very little existing literature have determined such characteristics using multiple approaches simultaneously in the retail store. Hence, the aim of this study was to identify, categorize, and rank the traits of retail stores in relation to customer satisfaction. To achieve this goal, the following objectives were formulated: (i) develop a potential list of attributes that affect customer satisfaction; (ii) identify the types of attributes (must-be, one-dimensional, attractive, and indifferent), based on customers' perceptions; and (iii) rank and weight the identified attributes. The findings of this study will help retail store service providers understand the relative importance of the attributes, based on the level of their impact on customer satisfaction.

## Literature review

### Retail trade

The economic revenue of a country highly depends on retail trade, as it represents approximately six percent of the gross domestic product (GDP) in the US and four percent of the GDP in the EU-28 [12]. The industry is currently growing at a high rate and has a significant number of investors.

### Customer satisfaction

Satisfied customers are becoming a measure of economic wellbeing for retailers, with many stores tying customer satisfaction scores to employee compensation, signifying that a good satisfaction score indicates good revenue [13]. The successful operation of retail stores requires building strong relationships between the customers and the employees [4], as satisfied customers are generally loyal customers, who are frequent shoppers and recommend the store to their friends [3]. Hence, it is important for retailers to determine the factors that affect customer satisfaction. Sachdeva and Goel [14] found that service providers are focusing on making their customers' in-store shopping experience entertaining and educational to win them over from online retailing.

### Service quality

The quality of the service provided is one of the major factors that affects retail store customers' satisfaction and gives stores a competitive advantage [6, 13]. Service quality can be defined

as the difference between the service that customers expect and that which they actually receive [15]. Paul et al. [6] found that customers are happy when they have a good purchase outcome that fulfills their goal of making hassle-free purchases; however, their study mainly focused on discovering the relationship between customer satisfaction and the service quality of the bank sector. Other similar studies also focused on customer satisfaction with the service quality of the bank sector [4,16, etc.]. Agnihotri *et al.* [17] found that a salesperson's responsiveness has a positive influence on customers' satisfaction. Tontini *et al.* [18] discussed five dimensions of the service quality of online retail store customers' satisfaction: speed of service, recovery of errors, reliability, easily accessed information, and the importance of feedback. Ibrahim *et al.* [19] found that competent and well-behaved employees positively affect customers' satisfaction. Brady and Cronin Jr. [20] divided service quality into three sub-qualities: the quality of interaction, the quality of the physical environment, and the quality of the outcome. The first sub-category includes the dimensions related to attitude, behavior, and expertise; the second sub-category includes the dimensions related to ambient conditions, design, and social factors; and the last sub-category includes the dimensions of waiting time, tangibles, and valence. Izogo and Ogba [21] discussed customer satisfaction based on the service quality dimensions of reliability, responsiveness, assurance, empathy, and tangibles in the context of the automobile repair services sector. Chen *et al.* [22] and Nadiri and Tumer [23] found five service quality elements that influence customer satisfaction, namely physical image, reliability, personal interactions, problem solving, and store policies. Chen *et al.* [7] discussed these dimensions while focusing on department stores, whereas Nadiri and Tumer [23] focused on the effects of local culture on these dimensions. In 1996, Dabholkar et al. [24] proposed a hierarchical structure of retail service qualities, using confirmatory factor analysis. Retail service quality was subdivided into physical aspects, reliability, personal interactions, problem solving, and policies. Physical aspects were divided into appearance and convenience; reliability was divided into promises and the fulfillment of them; personal interaction was divided into confidence-inspiring confidence and helpful. The degree of impact that the above-mentioned attributes have on customer satisfaction rarely has been studied, however.

## Kano model

Retailers need to know the importance of the attributes that contribute to customer satisfaction [25], but until 1984, when Professor Noriako Kano created the Kano model [26], it was difficult to assess them. Kano connects customer satisfaction with product quality and functionality [27, 28], and uses a questionnaire survey to portray customer satisfaction and dissatisfaction by a graphical representation. The horizontal axis of the graph represents the fulfillment of the functionality of a feature, and the vertical axis represents customer satisfaction that is due to the fulfillment of the functionality of that feature. Not all of the features of a product or service are required by the consumers at the same level, however, so Kano divided them into three ways that they affect customer satisfaction: must-be quality, one-dimensional quality, and attractive quality [29, 30]. According to Avikal *et al.* [28], there are two more dimensions of customer perceptions of products and/or service quality: indifferent quality and reverse quality. Many more researchers have discussed these five perspectives of product characteristics [27, 31, 32]. Fig 1 shows the two-dimensional Kano model [33, 34].

## Kano categories

Detailed descriptions of the six Kano categories that impact customer satisfactions are as follows.

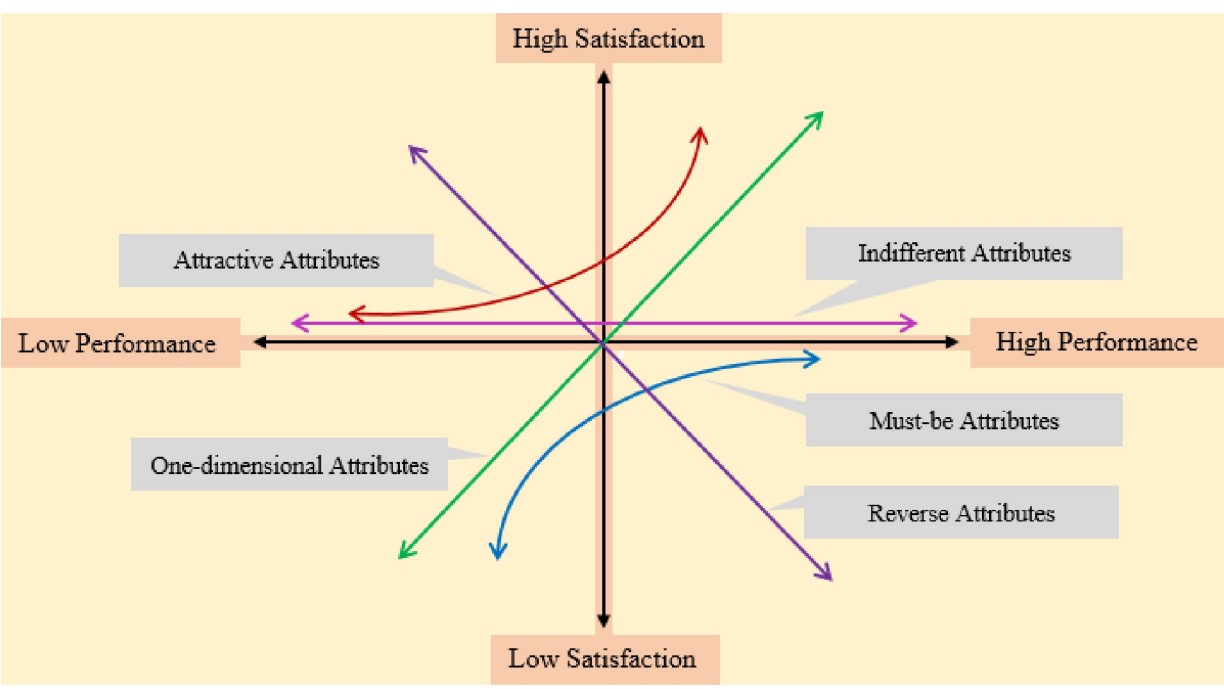

**Fig 1. Two-dimensional Kano model.**

- Must-be (M) qualities are the basic qualities [30] that determine whether a product or service is considered complete. They do not contribute to customer satisfaction, but their absence causes high dissatisfaction. They are sometimes called basic expectations or basic attributes [32, 35].

- One-dimensional (O) qualities follow the old belief that customer satisfaction is linear [32] —that it increases with the increase in the quality of a product/service and vice versa [30]. One-dimensional qualities are those in which the level of fulfillment is proportionate to how happy it makes the consumer, and vice versa. They are often called performance qualities [32] or linear attributes [35].

- Attractive (A) qualities are unexpected [31, 36] and increase customer satisfaction more than proportionately. Not having them does not make the consumers unhappy, as they were unforeseen [25, 37, 38]. These qualities are often called exciting attributes [35] or motivational attributes [32].

- Indifferent (I) qualities are those whose degree of fulfillment do not affect the level of customer satisfaction [28].

- Reverse (R) qualities are those qualities that make the consumer rather unhappy [28].

- Questionable (Q) attributes are those that indicate requirements with contradictory or confusing responses that need further investigation before being included in an analysis [37].

## Advantages of Kano model

The Kano model can help resolve financial constraints, but it is vital to identify the features that provide the maximum benefit with the minimum investment [30]. For example, if two or more service aspects need attention at the same time but resources and time are limited, the

Kano model can determine which service features influence customer satisfaction more and thus need immediate attention [30]. As time passes, qualities can change, i.e., attractive qualities can become one-dimensional qualities, and one-dimensional qualities can become must-be qualities [39]. Tracking the attributes of retail stores with the Kano model for consecutive time intervals provides the retailer with a competitive edge by revealing how customer choices are changing and which features are reclassified from one category to another.

Another major advantage of the Kano model is that can be used in different sectors of the economy [28, 32], along with other models and/or methods. Matzler and Hinterhuber [25] used this model successfully, in combination with quality function deployment, in product development projects. Lee *et al.* [40] combined its use with fuzzy mode in the product lifecycle management (PLM) system, and Garibay *et al.* [41] used it to evaluate the digital library. Hashim and Dawal [42] utilized this combination to improve ergonomic design, and Shahin [43] integrated the Kano model with failure mode and effect analysis (FMEA) to determine FMEA capabilities from the perspective of the customers, which had rarely been done before. Basfirinci and Mitra [44] used the Kano model, integrating it with SERVQUAL, to explore the quality of airline service. Darn *et al.* [45] used it to identify must- be, one dimensional, and attractive features while designing a website. It has rarely been used, however, to categorize and rank a comprehensive list of attributes of retail service facilities. After an exhaustive search through literature, the authors of this study found one article about the Kano model being used for a department store [7], and one article about it being used to determine online retail characteristics of customer satisfaction [18].

## Limitations of the Kano model

Similar to all other models, the Kano model has some limitations that researchers have been trying to overcome in different ways for many years. For example, Xu *et al.* [36] concluded that as a Kano model mostly focuses on customer satisfaction qualitatively, without considering the producers' capability, it is not suitable for engineering designs. They developed an analytical Kano model, known as the A-Kano model, that considers quantitative measurements as well. Another approach to improving the basic Kano model was performed by replacing product quality with experience quality, thus adding users' emotions to eventually increase product quality [27, 46]. After conducting a thorough review of the Kano literature, Ek and Cikis [27] concluded that the limitations of the Kano model can be explained by three points. The first one is the ambiguous meaning of the wording of the Kano questionnaire and Kano evaluation table. The second one occurs when the situation demands more than five Kano categories to properly justify the outcome. And the third one occurs when one or more of the five categories becomes redundant.

## Research methodology

This study followed a four-step methodology (Fig 2). The first step focused on the literature review, from which a list was developed of attributes of retail sectors that affect customer satisfaction. The second step focused on the development of the survey, which consisted of two types of questions: Kano questions and self-stated importance questions. Before the study began, the survey questions were reviewed and approved by the Ethics Committee of the Eastern Mediterranean University. The questionnaires were also tested through a pilot survey and modified accordingly. Next, the survey was distributed to the respondents, and 400 responses were collected. The last step focused on analyzing the data and interpreting the results.

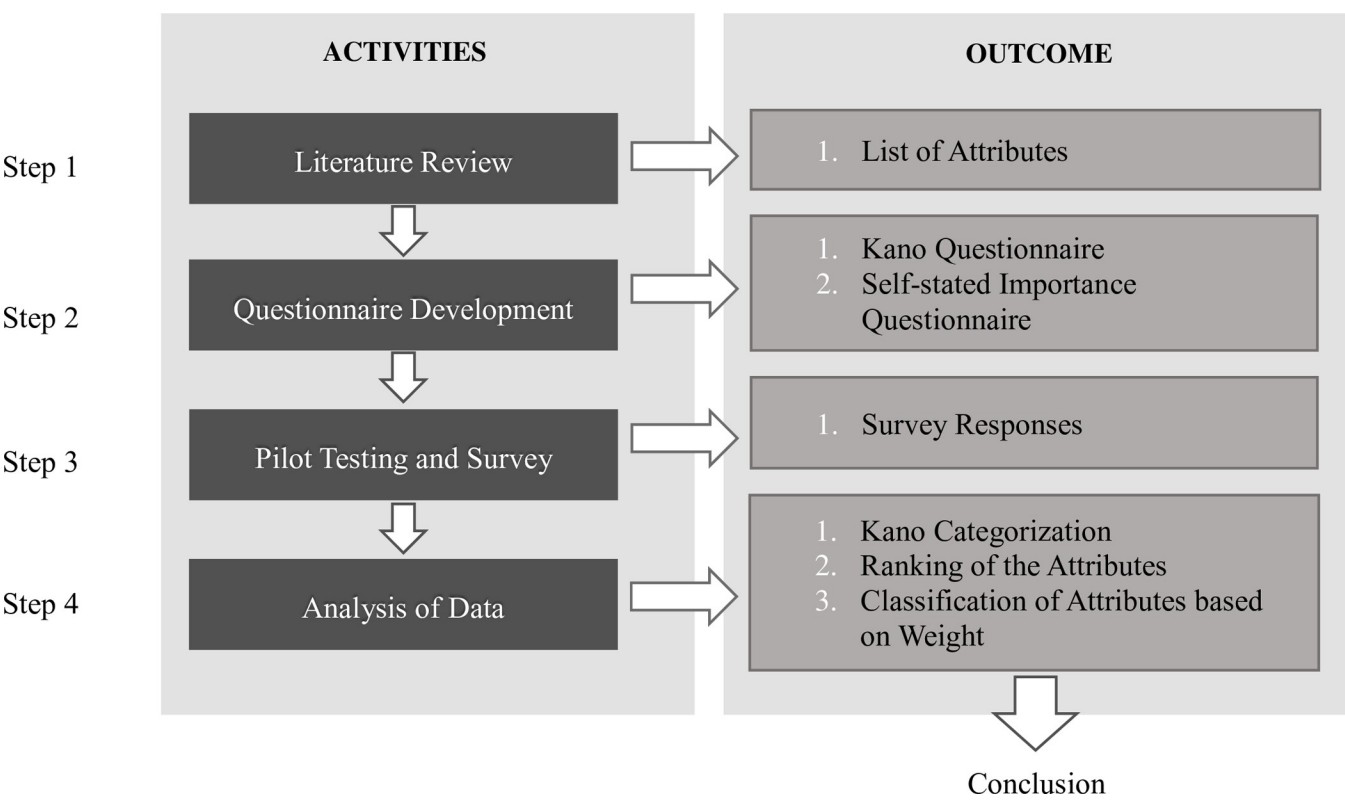

**Fig 2. Research methodology.**

## Survey development

This study aimed to define the attributes of a retail store that affect customer satisfaction. An extensive literature review was conducted, and a list was developed of twenty-eight attributes of retail stores that affect customers' satisfaction. The list is shown in Table 1.

As this study adopted the Kano model to evaluate customer satisfaction based on the attributes of a retail store, the questions had to be designed in a manner that would help determine whether the customer and the service provider assign the same level of importance to certain features. For this purpose, a pair of questions was developed for each attribute, which is the most appropriate way to determine the Kano categories for each feature of the service [22]. The first question of the pair was a functional question that evaluated the level of customer satisfaction when the requirement related to the feature was met, and the second question was a dysfunctional question that evaluated the level of customer satisfaction when the requirement was not met. Questions related to two attributes of retail stores are provided as a sample of the survey questions (functional, dysfunctional, and self-stated questions) in Table 2.

In parallel with the functional and dysfunctional questions, a self-stated important question was included for each attribute. A scale from 1 to 7, with 1 being not at all important and 7 being extremely important, was provided to determine the relative importance of each attribute to their satisfaction level. Survey was prepared such way so that it can be completed within 15 minutes. The survey questions for this study were reviewed by a committee of experienced faculty members Eastern Mediterranean University. The questions were approved and ensured that research ethics were properly applied.

**Table 1. List of attributes.**

| # | Attributes | # | Attributes |
|---|---|---|---|
| 1 | Having modern looking equipment and fixtures | 15 | Having employees who can provide prompt services |
| 2 | Having visually appealing physical facilities | 16 | Having well-informed employees who can inform about the exact timing of the service |
| 3 | Having visually appealing service materials | 17 | Having employees who will never be distracted while responding to a request |
| 4 | Having clean and convenient public areas | 18 | Having the ability to providing individual attention |
| 5 | Having a layout that makes searching for materials easier | 19 | Having employees with consistent courteous behavior |
| 6 | Having a layout that makes moving around in the store easier | 20 | Having employees who will be courteous over the phone |
| 7 | Having to do the work within previously promised time | 21 | Having the willingness to handle returns and exchanges |
| 8 | Providing service right at the previously promised time | 22 | Paying sincere interest in solving customers problems |
| 9 | Performing the service right at the first time | 23 | Having competent employees who can handle complaints directly and immediately |
| 10 | Having merchandise available when customer wants it | 24 | Offering high-quality merchandise |
| 11 | Insisting on error-free sales transactions and records | 25 | Providing plenty of convenient parking |
| 12 | Having knowledgeable employees who can answer questions | 26 | Having convenient operating hours |
| 13 | Having good-behaved employees who can instill confidence in customer | 27 | Accepting most major credit cards |
| 14 | Having the ability to ensure safety during transactions | 28 | Offering own credit card of the store |

## Pilot testing and survey distribution

A sampling of 525 young people who were regular customers of retail stores was selected to receive the survey. Caution was given not to include any minor in this study. More than 50% were from the age group of 21–24, only 3% were older than 33, and 70% of them had a monthly income less than 4000 TL. Before conducting the survey, the questionnaire was pilot tested with 30 individuals to evaluate the survey questions prior to a larger distribution. The questionnaires were revised, based on the results of the pilot test, to reflect the purpose of the study. The pilot testing needed one month to be completed. The major distribution of the survey took four months to be completed. Hence, the distribution of the survey took total of five months from the initial distribution of the survey till the start of the analysis. The survey responses were collected by paper. Total of 400 survey responses were collected with the response rate of 76%. In the first page of the survey, respondents were notified that participation in the survey was voluntarily and their written consent was collected.

## Analysis and results

### Kano categorization

Data based on the responses to the functional and dysfunctional questions for each attribute was collected and analyzed to categorize the attributes of retail stores, using the Kano evaluation table. A Kano evaluation table was established using the references of [37, 47, 48]. The format of a pair of questions and the Kano evaluation table are shown in Table 3.

**Table 2. Sample of survey questions.**

| Attribute 1: Having modern looking equipment | |
|---|---|
| 1a. If the store has modern-looking equipment, how would you feel? | 1. I like it that way<br>2. It must be that way<br>3. I am neutral<br>4. I can live with it that way<br>5. I dislike it that way |
| 1b. If the store doesn't have modern-looking equipment, how would you feel? | 1. I like it that way<br>2. It must be that way<br>3. I am neutral<br>4. I can live with it that way<br>5. I dislike it that way |
| 1c. How important it is for you that the store has modern-looking equipment? | Not at all important  Extremely important<br><br>1   2   3   4   5   6   7 |
| Attribute 2: Having visually appealing physical facilities | |
| 2a. If the store has visually appealing physical facilities, how would you feel? | 1. I like it that way<br>2. It must be that way<br>3. I am neutral4. I can live with it that way5. I dislike it that way |
| 2b. If the store doesn't have visually appealing physical facilities, how would you feel? | 1. I like it that way<br>2. It must be that way<br>3. I am neutral<br>4. I can live with it that way<br>5. I dislike it that way |
| 2c. How important it is for you that the store has visually appealing physical facilities? | Not at all important  Extremely important<br><br>1   2   3   4   5   6   7 |

Each response was placed into one of the six Kano categories, based on the evaluation model. A preliminary selection process was performed to exclude the attributes that received a significant number of questionable (Q) scores until the confusion regarding the question was cleared. Attributes with a significant number of reverse (R) scores were also excluded from the analysis initially, as they indicated that the respondents' perception of a particular attribute was the opposite of the retailers'. These attributes were included in the analysis only after switching the responses of the functional and dysfunctional questions. Once an attribute with a significant reverse score was identified, the responses were switched for all of the responses for that particular attribute, irrespective of whether the original response was reverse. Table 4 demonstrates the frequency of responses per category. The far-right column of Table 4 shows the Kano category that was assigned after all of the responses had been analyzed for each feature.

**Table 3. Kano evaluation table with corresponding response option for functional and dysfunctional questions.**

| Question type | | Dysfunctional: "If [the service] did not satisfy [requirement x], how would you feel?" | | | | |
|---|---|---|---|---|---|---|
| | Response options | I like it that way | It must be that way | I am neutral | I can live with it that way | I dislike it that way |
| Functional: "If [the service] satisfied [requirement x], how would you feel?" | I like it that way | Questionable | Attractive | Attractive | Attractive | One-dimensional |
| | It must be that way | Reverse | Indifferent | Indifferent | Indifferent | Must-be |
| | I am neutral | Reverse | Indifferent | Indifferent | Indifferent | Must-be |
| | I can live with it that way | Reverse | Indifferent | Indifferent | Indifferent | Must-be |
| | I dislike it that way | Reverse | Reverse | Reverse | Reverse | Questionable |

**Table 4. Kano category based on frequency of responses.**

| # | Attributes | Most Frequent Category | 2nd Most Frequent Category | 3rd Most Frequent Category | Kano Category |
|---|---|---|---|---|---|
| 2 | Having visually appealing physical facilities | M (199) | I (118) | O (39) | Must-be |
| 10 | Having merchandise available when customer wants it | M (128) | I (114) | O (82) | |
| 11 | Insisting on error-free sales transactions and records | M (147) | I (131) | O (58) | |
| 19 | Having employees with consistent courteous behavior | M (128) | I (109) | O (86) | |
| 27 | Accepting most major credit cards | M (114) | I (110) | O (90) | |
| 25 | Providing plenty of convenient parking | M (187) | I (124) | O (43) | |
| 1 | Having modern looking equipment and fixtures | O (117) | A (113) | I (108) | One-dimensional |
| 4 | Having clean and convenient public areas | O (125) | I (119) | A (80) | |
| 5 | Having a layout that makes searching for materials easier | O (152) | I (90) | A (73) | |
| 7 | Having to do the work within previously promised time | O (140) | I (117) | M (66) | |
| 13 | Having good-behaved employees who can instill confidence in customer | O (128) | I (118) | A (82) | |
| 17 | Having employees who will never be distracted while responding to a request | O (134) | I (110) | A (97) | |
| 21 | Having the willingness to handle returns and exchanges | O (153) | M (96) | I (78) | |
| 26 | Having convenient operating hours | O (132) | I (127) | A (83) | |
| 28 | Offering own credit card of the store | O (143) | I (103) | M (66) | |
| 6 | Having a layout that makes moving around in the store easier | A (155) | I (134) | O (60) | Attractive |
| 9 | Performing the service right at the first time | A (187) | I (140) | O (35) | |
| 14 | Having the ability to ensure safety during transactions | A (141) | I (129) | O (65) | |
| 18 | Having the ability to providing individual attention | A (137) | I (136) | O (56) | |
| 22 | Paying sincere interest in solving customers problems | A (146) | I (109) | O (77) | |
| 23 | Having competent employees who can handle complaints directly and immediately | A (175) | I (123) | O (51) | |
| 3 | Having visually appealing service materials | I (175) | A (91) | O (69) | Indifferent |
| 8 | Providing the service right at the previously promised time | I (129) | O (93) | A (89) | |
| 12 | Having knowledgeable employees who can answer questions | I (143) | A (81) | M (79) | |
| 15 | Having employees who can provide prompt services | I (157) | A (83) | O (77) | |
| 16 | Having well-informed employees who can inform about the exact timing of the service | I (175) | A (85) | O (68) | |
| 20 | Having employees who will be courteous over the phone | I (150) | O (92) | A (71) | |
| 24 | Offering high-quality merchandise | I (132) | A (99) | O (98) | |

Among the listed 28 attributes related to customer satisfaction in the retail sector, 6 were categorized as basic or must-be attributes: appealing physical facilities, merchandise that is available when the customer wants it, error-free sales transactions, having courteous employees, acceptance of most major credit cards, multiple convenient locations. It is worth noting again that according to the definition of the category, even though the inability to provide these facilities reduces customer satisfaction significantly, providing them does not increase it.

The second category, the one-dimensional or linear attribute, causes satisfaction that is proportional to the quality of the provided feature and improving them improves customer satisfaction at a constant rate. The attributes in this category are modern-looking equipment and fixtures, clean and convenient public areas, easy storage layout, performance of service when promised, well-behaved employees that boost the customer's confidence, employees who are focused on the task at hand, willingness to handle returns and exchanges, convenient operating hours, and a store credit card.

The third category of attributes, attractive and/or motivational, causes customers to be unexpectedly satisfied and instills loyalty to the store. These attributes include a store layout

that is easy to navigate, services that are performed correctly the first time, safe transactions, personalized attention, demonstrated sincere interest in solving customers' problems, and competent employees who can handle complaints directly and immediately.

The fourth category is the indifferent category, and it includes 7 of the 28 attributes: visually appealing service materials, providing the service correctly when promised, knowledgeable employees who can answer questions, employees who provide prompt services, well-informed employees who can provide accurate information about the exact timing of a service, employees who are courteous over the phone, and high quality merchandise Attributes from this category bring neither customer satisfaction nor dissatisfaction, thus have very little impact on overall customer satisfaction.

The categorizations are based on the maximum number of responses, which might or might not be significantly higher than the second highest frequent response for the respective attribute. For example, attribute 10, which is having merchandise available when the customer wants it, is categorized as a must-be attribute, based on the 128 most frequent responses; however, the second-highest frequent response for this attribute is "indifferent," with 114 responses. Similarly, even though 129 respondents (most frequent response for attribute 8) found that providing service at the previously promised time does not make any difference in their satisfaction level, 93 respondents (the second highest frequent response for this attribute) found this feature attractive. Hence, to consider the effect of all of the responses, multiple refined analyses of Kano responses were used and are presented in the following sections.

## Ranking of the attributes

Kano categories have a definite hierarchical rule that is based on the influence that attributes have on customers' satisfaction/dissatisfaction levels. For example, must-be is the most influential, followed by one-dimensional, attractive, and indifferent, which means that the indifferent category has the least number of influential attributes. Hence, it is important to acknowledge all of the responses when evaluating and categorizing the attributes. Timko's [49] modification of the Kano model that uses the total satisfaction index based on Kano responses was used for this. This method calculates better and worse values, using the following formulas, to understand the extent of customers' satisfaction and dissatisfaction with attributes [31, 47, 48, 50].

$$Better = \frac{A + O}{A + O + M + I} \tag{1}$$

$$Worse = \frac{O + M}{(A + O + M + I)x(-1)} \tag{2}$$

The difference between the better and worse values is known as the total satisfaction index, and the attributes can be ranked, based on the calculated values of the total satisfaction index. Negative values of the total satisfaction index indicate that failing to fulfill a specific attribute will cause dissatisfaction, and positive values indicate that fulfilling a specific attribute will cause satisfaction. In addition, the higher values yield the greater impacts.

As the attributes belonging to the one-dimensional category have linear relationships with customer satisfaction, they should be fulfilled at a minimum level. As the attributes of the one-dimensional and indifferent categories have a low value on the total satisfaction index, it is not useful to rank them. As a result, this section of the analysis focuses mainly on attributes from the must-be and attractive categories. The results of the calculations of the total satisfaction index are shown in Table 5.

**Table 5. Ranking of attributes from must-be and attractive categories.**

| # | Attributes | Better = (A+O)/ (A+O +M+I) | Worse = -(O+M)/ (A+O +M+I) | Total Satisfaction Index | Ranking |
|---|---|---|---|---|---|
| | Must-be Attributes | | | | |
| 2 | Having visually appealing physical facilities | 0.18 | -0.61 | -0.43 | 1 |
| 25 | Providing plenty of convenient parking | 0.19 | -0.6 | -0.41 | 2 |
| 11 | Insisting on error-free sales transactions and records | 0.27 | -0.54 | -0.26 | 3 |
| 19 | Having employees with consistent courteous behavior | 0.37 | -0.57 | -0.2 | 4 |
| 10 | Having merchandise available when customer wants it | 0.36 | -0.55 | -0.19 | 5 |
| 27 | Accepting most major credit cards | 0.41 | -0.54 | -0.13 | 6 |
| | Attractive Attributes | | | | |
| 9 | Performing the service right at the first time | 0.59 | -0.14 | 0.45 | 1 |
| 23 | Having competent employees who can handle complaints directly and immediately | 0.59 | -0.22 | 0.38 | 2 |
| 6 | Having a layout that makes moving around in the store easier | 0.55 | -0.26 | 0.3 | 3 |
| 22 | Paying sincere interest in solving customers problems | 0.6 | -0.32 | 0.28 | 4 |
| 14 | Having the ability to ensure safety during transactions | 0.55 | -0.28 | 0.26 | 5 |
| 18 | Having the ability to providing individual attention | 0.52 | -0.27 | 0.25 | 6 |

It is important for service providers with limited resources to know which attributes to focus on. The attributes that have a negative total satisfaction index (closer to -1) are the ones that customers expect from retail stores. The first attribute that customers want in order to be satisfied are physical facilities that are visually appealing (with -0.43 satisfaction index). Plenty of convenient parking spaces is the second attribute within the must-be category (with -0.41 satisfaction index).

On the other hand, the attributes with a higher total satisfaction index (closer to 1) have a stronger influence on customers' satisfaction. The customers' responses made it clear that performing a service correctly the first time (0.45 satisfaction index) is the most influential attribute within the attractive category, followed by the employees' ability to handle complaints immediately and directly (0.38 satisfaction index). As depicted in Table 5, allocating resources to attributes, based on their ranking and impact level, will help retail service providers attract customers from their competitors.

## Weighting of the attributes

Another way to analyze a Kano questionnaire is by a graphical representation of the categories that is based on the weighted functional and dysfunctional averages [48, 51]. Accordingly, the responses to the questionnaire were identified through the conversion numbers shown in Table 6 to determine the functional and dysfunctional weighted averages.

When the attributes are categorized based on the maximum number of respondents' perceptions, the second, third, and subsequent responses are neglected. Therefore, instead of using the most frequent response to categorize attributes, every response given by the participants can be considered by scoring the responses of the functional and dysfunctional conditions. Then, the functional and dysfunctional weighted average is determined by using the

**Table 6. Conversion of responses into numbers.**

| Customer Requirement | 1. Like | 2. Must be | 3. Neutral | 4. Live with | 5. Dislike |
|---|---|---|---|---|---|
| Functional | 4 | 2 | 0 | -1 | -2 |
| Dysfunctional | -2 | -1 | 0 | 2 | 4 |

equations provided below:

$$Dysfunctional\ weighted\ average, X = \frac{\sum(Dysfunctional\ Items)}{N} \tag{3}$$

$$Functional\ weighted\ average, Y = \frac{\sum(Functional\ Items)}{N} \tag{4}$$

Here, N represents the sample size.

The calculated values for the functional and dysfunctional weighted averages are shown in Table 7. For each attribute, the functional and dysfunctional numbers can be converted into two coordinates (X, Y) of a two-dimensional quadrant system.

Service providers are not interested in categorizing attributes; they want to know which attributes should be given the highest priority and which ones can be tended to later. This can be addressed by considering the importance level of attributes while presenting the average

**Table 7. Functional and dysfunctional weighted average, along with importance level.**

| # | Attributes | Functional Weighted Average (Y) | Dysfunctional Weighted Average (X) | Importance Level |
|---|---|---|---|---|
| 1 | Having modern looking equipment and fixtures | 2.77 | 1.56 | 5.15 |
| 2 | Having visually appealing physical facilities | 1.10 | 2.36 | 6.03 |
| 3 | Having visually appealing service materials | 2.20 | 2.62 | 3.84 |
| 4 | Having clean and convenient public areas | 2.57 | 2.76 | 5.02 |
| 5 | Having a layout that makes searching for materials easier | 2.77 | 2.44 | 5.24 |
| 6 | Having a layout that makes moving around in the store easier | 2.48 | 2.58 | 4.90 |
| 7 | Having to do the work within previously promised time | 2.47 | 2.44 | 5.06 |
| 8 | Providing service right at the previously promised time | 2.39 | 2.65 | 3.93 |
| 9 | Performing the service right at the first time | 2.65 | 1.85 | 4.90 |
| 10 | Having merchandise available when customer wants it | 1.77 | 1.64 | 5.69 |
| 11 | Insisting on error-free sales transactions and records | 1.44 | 2.16 | 5.62 |
| 12 | Having knowledgeable employees who can answer questions | 2.07 | 2.62 | 3.86 |
| 13 | Having good-behaved employees who can instill confidence in customer | 2.52 | 2.16 | 5.06 |
| 14 | Having the ability to ensure safety during transactions | 2.55 | 2.23 | 4.89 |
| 15 | Having employees who can provide prompt services | 2.16 | 2.21 | 3.98 |
| 16 | Having well-informed employees who can inform about the exact timing of the service | 2.03 | 2.76 | 3.81 |
| 17 | Having employees who will never be distracted while responding to a request | 2.72 | 2.29 | 5.01 |
| 18 | Having the ability to providing individual attention | 2.27 | 1.84 | 4.79 |
| 19 | Having employees with consistent courteous behavior | 1.75 | 2.25 | 5.63 |
| 20 | Having employees who will be courteous over the phone | 2.14 | 2.41 | 3.81 |
| 21 | Having the willingness to handle returns and exchanges | 2.40 | 2.61 | 5.13 |
| 22 | Paying sincere interest in solving customers problems | 2.61 | 2.69 | 5.12 |
| 23 | Having competent employees who can handle complaints directly and immediately | 2.65 | 2.19 | 5.34 |
| 24 | Offering high-quality merchandise | 2.06 | 2.26 | 5.53 |
| 25 | Providing plenty of convenient parking | 2.26 | 3.12 | 5.65 |
| 26 | Having convenient operating hours | 3.24 | 0.86 | 5.03 |
| 27 | Accepting most major credit cards | 0.52 | 1.48 | 5.68 |
| 28 | Offering own credit card of the store | 1.13 | 0.43 | 3.84 |

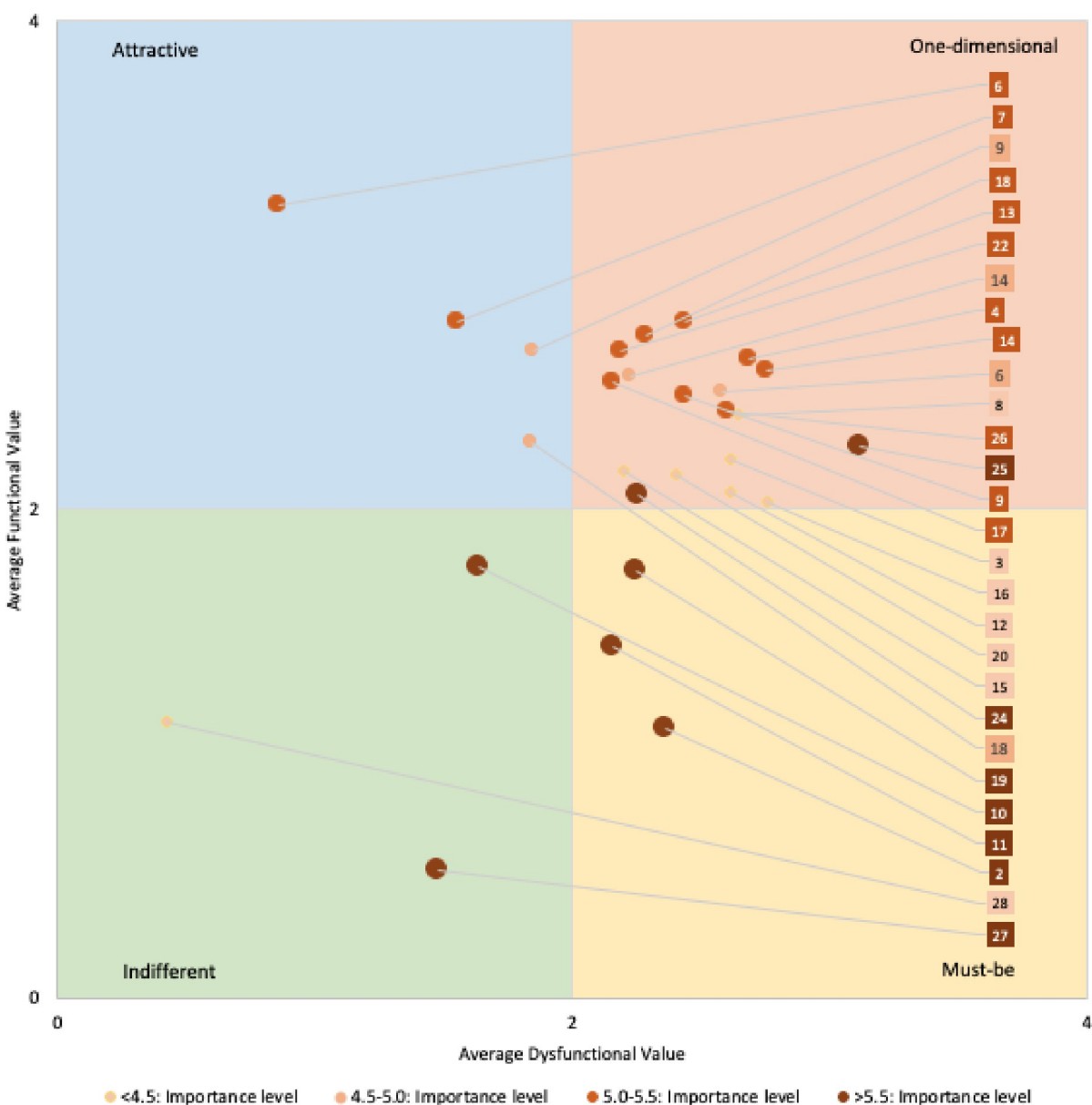

**Fig 3. Graphical representation of Kano categories, along with average importance level.**

functional and dysfunctional weighted scores. Fig 3 was developed by taking the dysfunctional weighted average along the x-axis and the functional weighted average along the y-axis to find the level of importance for each attribute. The graph shows the relative position and importance level of each attribute in the Kano category.

The Fig 3 graph has four quadrants that represent four Kano categories. The top left quadrant holds the attractive qualities, the top right quadrant holds the one-dimensional qualities, the bottom right quadrant holds the must-be qualities, and the bottom left quadrant holds the indifferent qualities. Based on the importance scale (Table 7), the top seven attributes with a score greater than 5.5 were identified on the graph (Fig 3). It was found that among these seven attributes, three of them (well-behaved employees, error-free sales transactions, and visually appealing physical facilities) fell into the must-be quadrant. Two of them (easy-to-

navigate store layout and delivering services when promised) fell into the attractive quadrant, and the other two (having merchandise available when customer wants it and accepting most major cards) fell into the indifferent quadrant. Even though several attributes fell into the quadrant of one-dimensional categories, the importance score reveals which needs immediate attention.

## Conclusion

To gain a competitive advantage, owners and/or managers in the retail sector need to know the attributes that are important to their customers. This study aimed not only to determine such attributes, but also to rank them so that the retailers could determine which ones were the most important. To fulfill the purpose of the study, a comprehensive literature review was conducted, 28 retail store attributes were determined, a Kano questionnaire was distributed, and Kano model was used to evaluate the attributes. It was explicated in this research that according to the respondents, the fulfillment of different attributes affects customer satisfaction at different levels. Results also indicated that a retail store should have visually appealing facilities in order to sustain satisfied customers, as this is a must-be attribute, the absence of which would significantly decrease customer satisfaction. An easy-to-navigate layout is an attribute from the attractive category that has a proportional relationship with customer satisfaction, but its absence does not cause customer dissatisfaction. Attributes such as providing clean rest areas have a proportional relationship with customer satisfaction. Based on the self-stated importance questionnaire, it was found that most of the attributes with a high level of importance are from the must-be and one-dimensional categories. The findings of this study will help service providers become more knowledgeable about the relative importance of the attributes of retail stores and will enable them to evaluate the impact of their current practices on customer satisfaction levels.

This study demonstrated the evaluation of customer satisfaction in the retail sector, using the Kano model. It should be noted, however, that the results of this study are mainly representative of young customers, as that was the target group of the questionnaire, and they should not be generalized for customers from other age groups.

The inclusion of customers from several other age groups as target participants will widen the scope of this research. Similar studies could be performed in different parts of the world with the different economic conditions of the people which will help in identifying the connection between income level and customer satisfaction characteristics for the retail sector if there is any. Similarly, this research could be replicated in sectors other than retail, for example, supply sector, to determine the characteristics to improve for better performance and satisfaction.

## Supporting information

**S1 Data.**
(XLSX)

**S1 Appendix.**
(DOCX)

## Author Contributions

**Conceptualization:** Sharareh Kermanshachi, Halil Nadiri.

**Data curation:** Thahomina Jahan Nipa.

**Formal analysis:** Sharareh Kermanshachi, Halil Nadiri.

**Investigation:** Sharareh Kermanshachi, Thahomina Jahan Nipa.

**Methodology:** Sharareh Kermanshachi.

**Supervision:** Sharareh Kermanshachi, Halil Nadiri.

**Writing – original draft:** Thahomina Jahan Nipa.

**Writing – review & editing:** Sharareh Kermanshachi, Thahomina Jahan Nipa.

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
