## [Decision Letter · Decision Letter 0]

28 May 2021

PONE-D-21-07256

Service Quality Assessment and Enhancement Using Kano Model

PLOS ONE

Dear Dr. Kermanshachi,

Thank you for submitting your manuscript to PLOS ONE. After careful consideration, we feel that it has merit but does not fully meet PLOS ONE’s publication criteria as it currently stands. Therefore, we invite you to submit a revised version of the manuscript that addresses the points raised during the review process.

We recommend that it should be revised taking into account the changes requested by the reviewers. Since the requested changes include Major Revision, the revised manuscript will undergo the next round of review by the same reviewers.

We look forward to receiving your revised manuscript.

Kind regards,

Baogui Xin, Ph.D.

Academic Editor

PLOS ONE

Journal Requirements:

Reviewers' comments:

Reviewer's Responses to Questions

**Comments to the Author**

1. Is the manuscript technically sound, and do the data support the conclusions?

Reviewer #1: Partly

Reviewer #2: Yes

2. Has the statistical analysis been performed appropriately and rigorously? 

Reviewer #1: Yes

Reviewer #2: Yes

3. Have the authors made all data underlying the findings in their manuscript fully available?

Reviewer #1: Yes

Reviewer #2: Yes

4. Is the manuscript presented in an intelligible fashion and written in standard English?

Reviewer #1: Yes

Reviewer #2: Yes

5. Review Comments to the Author

Reviewer #1: There should be better explanation of literature gap, more details about the way that the questionnaire was conducted (time frame, and other similar details). The questionnaire should be part of the paper... There are no limitations of this research as well as suggestions for future research.

Reviewer #2: The goal of the article is clear.

1. An 'Abstract" should be added to recapitulate the main purpose, principal arguments, key observations and policy implications of the study.

2. There are minor linguistic and grammatical inconsistencies in the text. The manuscript would benefit from thorough editing for improvement of English.

6. PLOS authors have the option to publish the peer review history of their article (what does this mean?). If published, this will include your full peer review and any attached files.

Reviewer #1: No

Reviewer #2: No

---

## [Author Response · Author response to Decision Letter 0]

26 Nov 2021

Response to Reviewers

Response:

Thank you for your comment. The manuscript is formatted and named according to the PLOS ONE’s style requirements. 

Response:

Thank you for your comment. The questionnaire is included as supporting information. We also included minimal anonymized data set as supporting information necessary to replicating the analysis.

Response:

Thank you for your comment. We have collected written consent in the survey while explaining the survey instruction. Screenshot of the survey instruction along with the process of collecting consent is provided below. 

Response:

Thank you for your comment. We included minimal anonymized data set as supporting information necessary to replicating the analysis.

Response:

Thank you for your comment. There are no ethical or legal restrictions on sharing a de-identified data set. We included minimal anonymized data set as supporting information necessary to replicating the analysis. For more information please provide a data request to Sharareh Kermanshachi (sharareh.kermanshachi@uta.edu).

Response:

Thank you for your comment. We included minimal anonymized data set as supporting information necessary to replicating the analysis.

Reviewers' comments:

Reviewer's Responses to Questions

Comments to the Author

1. Is the manuscript technically sound, and do the data support the conclusions?

Reviewer #1: Partly

Reviewer #2: Yes

2. Has the statistical analysis been performed appropriately and rigorously? 

Reviewer #1: Yes

Reviewer #2: Yes

3. Have the authors made all data underlying the findings in their manuscript fully available?

Reviewer #1: Yes

Reviewer #2: Yes

4. Is the manuscript presented in an intelligible fashion and written in standard English?

Reviewer #1: Yes

Reviewer #2: Yes

5. Review Comments to the Author

Reviewer #1: There should be better explanation of literature gap, more details about the way that the questionnaire was conducted (time frame, and other similar details). The questionnaire should be part of the paper... There are no limitations of this research as well as suggestions for future research.

Response:

We are grateful to have this comment. This comment provides us opportunity to improve our work. Following modifications are made to address the above comments.

Page 3:

To enjoy an advantage over its competitors, a service provider must know the attributes that contribute to their customers’ satisfaction. Several studies have employed various research approaches to identifying customer satisfaction characteristics in different sectors as well as the retail sector. Kim et al. (2020) studied customer equity and customer satisfaction in traditional and new retail formats using regression analysis. In another study, Nicod et al (2020) found that providing proactive training will increase the sales value per customer but will not enhance customer satisfaction. Veloso et al (2017) established using a multi-level and hierarchical model that the customer satisfaction and service perceived quality has a significant correlation among them with respect to the retail industry. The above-mentioned studies have a limited scope in a prioritized list of characteristics to focus on when enhancing customer satisfaction in the retail store. Moreover, very little existing literature have determined such characteristics using multiple approaches simultaneously in the retail store. Hence, the aim of this study was to identify, categorize, and rank the traits of retail stores in relation to customer satisfaction. To achieve this goal, the following objectives were formulated: (1) develop a potential list of attributes that affect customer satisfaction; (2) identify the types of attributes (must-be, one-dimensional, attractive, and indifferent), based on customers’ perceptions; and (3) rank and weight the identified attributes. The findings of this study will help retail store service providers understand the relative importance of the attributes, based on the level of their impact on customer satisfaction.

Page 12:

In parallel with the functional and dysfunctional questions, a self-stated important question was included for each attribute. A scale from 1 to 7, with 1 being not at all important and 7 being extremely important, was provided to determine the relative importance of each attribute to their satisfaction level. Survey was prepared such way so that it can be completed within 15 minutes. The survey questions for this study were reviewed by a committee of experienced faculty members Eastern Mediterranean University. The questions were approved and ensured that research ethics were properly applied.

Page 22:

Limitation

This study demonstrated the evaluation of customer satisfaction in the retail sector, using the Kano model. It should be noted, however, that the results of this study are mainly representative of young customers, as that was the target group of the questionnaire, and they should not be generalized for customers from other age groups.

Suggestions for future research

The inclusion of customers from several other age groups as target participants will widen the scope of this research. Similar studies could be performed in different parts of the world with the different economic conditions of the people which will help in identifying the connection between income level and customer satisfaction characteristics for the retail sector if there is any. Similarly, this research could be replicated in sectors other than retail, for example, supply sector, to determine the characteristics to improve for better performance and satisfaction. 

Survey instruction mentioning time required to complete the survey.

Reviewer #2: The goal of the article is clear.

1. An 'Abstract" should be added to recapitulate the main purpose, principal arguments, key observations and policy implications of the study.

Response:

Thank you for your comment. Following changes are made to address this comment. 

Page 3:

Abstract

Success in the retail sector is highly dependent on customer satisfaction. Maintaining a competitive edge depends upon the service providers knowing and enacting what is important to their customers. Multiple studies have employed various research approaches to identifying characteristics of customer satisfaction in different sectors as well as retail sector. However, very few have determined such characteristics using multiple approaches simultaneously in the retail store. This study aims to identify, categorize, and rank the retail store attributes, based on their effects on customer satisfaction. A survey focusing on retail store characteristics that impact customer satisfaction was developed and distributed. Over 400 responses were collected and evaluated, using the Kano model. Results showed that visually appealing facilities and error-free transactions are of prime importance to customers. They are taken for granted, but their absence plays a significant role in customer dissatisfaction. An easy-to-navigate store layout and readily available service increase customer satisfaction, but their absence doesn’t decrease customer satisfaction. Clean public areas and modern-looking equipment are important, and improvements to them increase customer satisfaction at a proportional rate. The findings of this study will assist service providers in realizing the relative importance of the attributes of retail stores and in evaluating the impacts of their current practices on customer satisfaction levels. Such importance will help retail sector policy makers in mandating policies focusing on must-have attributes to preserve customer satisfaction. 

2. There are minor linguistic and grammatical inconsistencies in the text. The manuscript would benefit from thorough editing for improvement of English.

Response:

We are grateful to have this comment. We have thoroughly read our manuscript and improved our English.

---

## [Decision Letter · Decision Letter 1]

26 Jan 2022

PONE-D-21-07256R1Service Quality Assessment and Enhancement Using Kano ModelPLOS ONE

Dear Dr. Kermanshachi,

Thank you for submitting your manuscript to PLOS ONE. After careful consideration, we feel that it has merit but does not fully meet PLOS ONE’s publication criteria as it currently stands. Therefore, we invite you to submit a revised version of the manuscript that addresses the points raised during the review process.

I recommend that it should be revised taking into account the changes requested by Reviewers. I would like to give you a chance to revise your manuscript. To speed the review process, the manuscript will only be reviewed by the Academic Editor in the next round.

We look forward to receiving your revised manuscript.

Kind regards,

Baogui Xin, Ph.D.

Academic Editor

PLOS ONE

Reviewers' comments:

Reviewer's Responses to Questions

**Comments to the Author**

1. If the authors have adequately addressed your comments raised in a previous round of review and you feel that this manuscript is now acceptable for publication, you may indicate that here to bypass the “Comments to the Author” section, enter your conflict of interest statement in the “Confidential to Editor” section, and submit your "Accept" recommendation.

Reviewer #1: (No Response)

2. Is the manuscript technically sound, and do the data support the conclusions?

Reviewer #1: Yes

3. Has the statistical analysis been performed appropriately and rigorously? 

Reviewer #1: Yes

4. Have the authors made all data underlying the findings in their manuscript fully available?

Reviewer #1: Yes

5. Is the manuscript presented in an intelligible fashion and written in standard English?

Reviewer #1: Yes

6. Review Comments to the Author

Reviewer #1: Dear Authors,

Thank you for your improvements and comments. There are a couple of minor changes that should be made:

- questionnaire should be a part of the document (provided in anex part)

- when it is about the time frame, there should be described how long did the whole process of collecting data lasted (from distributing the questionnaire to the begining of the analysis) not the time frame needed to fill the questionnaire.

- additional information on how much questionnaries were distributed, what is the response level (percentage) could be usefull...

- Limitations and Suggestions for future research should be part of the Conclusion.

7. PLOS authors have the option to publish the peer review history of their article (what does this mean?). If published, this will include your full peer review and any attached files.

Reviewer #1: No

---

## [Author Response · Author response to Decision Letter 1]

4 Feb 2022

Response to Reviewers

Reviewers' comments:

Reviewer's Responses to Questions

Comments to the Author

1. If the authors have adequately addressed your comments raised in a previous round of review and you feel that this manuscript is now acceptable for publication, you may indicate that here to bypass the “Comments to the Author” section, enter your conflict-of-interest statement in the “Confidential to Editor” section, and submit your "Accept" recommendation.

Reviewer #1: (No Response)

2. Is the manuscript technically sound, and do the data support the conclusions?

Reviewer #1: Yes

3. Has the statistical analysis been performed appropriately and rigorously? 

Reviewer #1: Yes

4. Have the authors made all data underlying the findings in their manuscript fully available?

Reviewer #1: Yes

5. Is the manuscript presented in an intelligible fashion and written in standard English?

Reviewer #1: Yes

6. Review Comments to the Author

Reviewer #1: Dear Authors,

Thank you for your improvements and comments. There are a couple of minor changes that should be made:

- questionnaire should be a part of the document (provided in anex part)

Response:

Thank you very much for your comment. Questionnaire is added to the appendix at the end of the manuscript document. 

- when it is about the time frame, there should be described how long did the whole process of collecting data lasted (from distributing the questionnaire to the beginning of the analysis) not the time frame needed to fill the questionnaire.

Response:

Thank you for your valuable comment. The distribution of the survey took total of five months from the initial distribution of the survey till the start of the analysis. This information is added to the manuscript. The changes that are made to add this information in the manuscript is shown below.

Page 13:

Pilot testing and survey distribution

A sampling of 525 young people who were regular customers of retail stores was selected to receive the survey. Caution was given not to include any minor in this study. More than 50% were from the age group of 21-24, only 3% were older than 33, and 70% of them had a monthly income less than 4000 TL. Before conducting the survey, the questionnaire was pilot tested with 30 individuals to evaluate the survey questions prior to a larger distribution. The questionnaires were revised, based on the results of the pilot test, to reflect the purpose of the study. The pilot testing needed one month to be completed. The major distribution of the survey took four months to be completed. Hence, the distribution of the survey took total of five months from the initial distribution of the survey till the start of the analysis. The survey responses were collected by paper. Total of 400 survey responses were collected with the response rate of 76%. In the first page of the survey, respondents were notified that participation in the survey was voluntarily and their written consent was collected. 

- additional information on how much questionaries were distributed, what is the response level (percentage) could be useful...

Response:

Thank you for your insightful comment. We distributed 525 surveys and collected 400 survey response. This resulted in 76% response rate. This information is also added in the manuscript. Following changes are made in the manuscript to add this information.

Page 13:

Pilot testing and survey distribution

A sampling of 525 young people who were regular customers of retail stores was selected to receive the survey. Caution was given not to include any minor in this study. More than 50% were from the age group of 21-24, only 3% were older than 33, and 70% of them had a monthly income less than 4000 TL. Before conducting the survey, the questionnaire was pilot tested with 30 individuals to evaluate the survey questions prior to a larger distribution. The questionnaires were revised, based on the results of the pilot test, to reflect the purpose of the study. The pilot testing needed one month to be completed. The major distribution of the survey took four months to be completed. Hence, the distribution of the survey took total of five months from the initial distribution of the survey till the start of the analysis. The survey responses were collected by paper. Total of 400 survey responses were collected with the response rate of 76%. In the first page of the survey, respondents were notified that participation in the survey was voluntarily and their written consent was collected. 

- Limitations and Suggestions for future research should be part of the Conclusion.

Response:

Thank you for your valuable comment. The limitations and suggestions are included as part of the Conclusion instead of presenting them separately. Following changes are made in the manuscript.

Conclusion

To gain a competitive advantage, owners and/or managers in the retail sector need to know the attributes that are important to their customers. This study aimed not only to determine such attributes, but also to rank them so that the retailers could determine which ones were the most important. To fulfill the purpose of the study, a comprehensive literature review was conducted, 28 retail store attributes were determined, a Kano questionnaire was distributed, and Kano model was used to evaluate the attributes. It was explicated in this research that according to the respondents, the fulfillment of different attributes affects customer satisfaction at different levels. Results also indicated that a retail store should have visually appealing facilities in order to sustain satisfied customers, as this is a must-be attribute, the absence of which would significantly decrease customer satisfaction. An easy-to-navigate layout is an attribute from the attractive category that has a proportional relationship with customer satisfaction, but its absence does not cause customer dissatisfaction. Attributes such as providing clean rest areas have a proportional relationship with customer satisfaction. Based on the self-stated importance questionnaire, it was found that most of the attributes with a high level of importance are from the must-be and one-dimensional categories. The findings of this study will help service providers become more knowledgeable about the relative importance of the attributes of retail stores and will enable them to evaluate the impact of their current practices on customer satisfaction levels.

This study demonstrated the evaluation of customer satisfaction in the retail sector, using the Kano model. It should be noted, however, that the results of this study are mainly representative of young customers, as that was the target group of the questionnaire, and they should not be generalized for customers from other age groups.

The inclusion of customers from several other age groups as target participants will widen the scope of this research. Similar studies could be performed in different parts of the world with the different economic conditions of the people which will help in identifying the connection between income level and customer satisfaction characteristics for the retail sector if there is any. Similarly, this research could be replicated in sectors other than retail, for example, supply sector, to determine the characteristics to improve for better performance and satisfaction. 

7. PLOS authors have the option to publish the peer review history of their article (what does this mean?). If published, this will include your full peer review and any attached files.

Do you want your identity to be public for this peer review? For information about this choice, including consent withdrawal, please see our Privacy Policy.

Reviewer #1: No

---

## [Editor Report · Decision Letter 2]

11 Feb 2022

Service Quality Assessment and Enhancement Using Kano Model

PONE-D-21-07256R2

Dear Dr. Kermanshachi,

We’re pleased to inform you that your manuscript has been judged scientifically suitable for publication and will be formally accepted for publication once it meets all outstanding technical requirements.

Kind regards,

Baogui Xin, Ph.D.

Academic Editor

PLOS ONE
---

## [Editor Report · Acceptance letter]

18 Feb 2022

PONE-D-21-07256R2 

Service Quality Assessment and Enhancement Using Kano Model 

Dear Dr. Kermanshachi:

I'm pleased to inform you that your manuscript has been deemed suitable for publication in PLOS ONE. Congratulations! Your manuscript is now with our production department. 

Kind regards, 

on behalf of

Professor Baogui Xin 

Academic Editor

PLOS ONE